# Dirichlet Graph Variational Autoencoder

**Jia Li**[1], **Jianwei Yu**[1], **Jiajin Li**[1], **Honglei Zhang**[3], **Kangfei Zhao**[1],
**Yu Rong**[2], **Hong Cheng**[1], **Junzhou Huang**[2]
[1] The Chinese University of Hong Kong
[2] Tencent AI Lab
[3] Georgia Institute of Technology
{lijia,jwyu,jjli,kfzhao,hcheng}@se.cuhk.edu.hk, zhanghonglei@gatech.edu
yu.rong@hotmail.com, jzhuang@uta.edu

## Abstract

Graph Neural Networks (GNNs) and Variational Autoencoders (VAEs) have been
widely used in modeling and generating graphs with latent factors. However, there
is no clear explanation of what these latent factors are and why they perform well.
In this work, we present Dirichlet Graph Variational Autoencoder (DGVAE) with
graph cluster memberships as latent factors. Our study connects VAEs based graph
generation and *balanced graph cut*, and provides a new way to understand and
improve the internal mechanism of VAEs based graph generation. Specifically,
we first interpret the reconstruction term of DGVAE as *balanced graph cut* in a
principled way. Furthermore, motivated by the *low pass* characteristics in *balanced graph cut*, we propose a new variant of GNN named Heatts to encode the
input graph into cluster memberships. Heatts utilizes the Taylor series for fast
computation of heat kernels and has better *low pass* characteristics than Graph
Convolutional Networks (GCN). Through experiments on graph generation and
graph clustering, we demonstrate the effectiveness of our proposed framework.

## 1 Introduction

Since the introduction of Graph Neural Networks (GNNs) [18, 4, 6] and Variational Autoencoders
(VAEs) [16], many studies [19, 21, 8] have used GNNs and VAEs (GVAEs) to generate realistic
graphs with latent factors. The latent factors are usually formulated as Gaussian variables and comply
with some predefined prior distributions, e.g., isotropic Gaussian with diagonal covariance. Although
some encouraging progress has been achieved, there is still little insight into the internal operation
and behaviour of these complex models, or what intrinsic information the latent factors have captured
with respect to the input graph. Inspired by the recent development of variational autoencoder topic
model [27, 3] in text generation, in this work, we propose to formulate the latent factors in GVAEs as
graph cluster memberships, analogous to topics in text generation.

Indeed, it is not a new idea to generate graphs via exploiting certain structures (i.e., clusters or
subgraphs). JT-VAE [14] proposes to generate molecular graphs in two phases, in which it first
decomposes the whole molecule into subgraphs or clusters of atoms, e.g., rings or bonds. It then
utilizes these clusters to assemble a coherent molecular graph. However, JT-VAE relies on tree
decomposition algorithms tailored for molecules, which is hard to generalize to non-molecular graph
generation. In this paper, we propose a novel Dirichlet Graph Variational Audoencoder (DGVAE) to
automatically encode the cluster decomposition in latent factors by replacing node-wise Gaussian
variables with Dirichlet distributions, where the latent factors can be taken as cluster memberships.
Under this framework, we provide a transparent interpretation of the de facto reconstruction term
of Evidence Lower Bound (ELBO) in VAEs, which is in effect equivalent to adopting the spectral

relaxed *balanced graph cut* [28]. In this vein, it further sheds light on the notable successful graph reconstruction performance.

Inspired by the connection between DGVAE and the spectral relaxed *balanced graph cut*, we refer to the literature of *balanced graph cut* to understand and improve DGVAE. Due to the hardness of *balanced graph cut*, previous solutions rely on spectral relaxation [25, 9] to solve the cut problem. Among them, the most popular approach is *spectral clustering*, which requires the computation of the first $K$ eigenvectors of the Laplacian matrix $L$, or *low pass* in spectral domain [25, 28]. Motivated by this *low pass* characteristics of *spectral clustering*, we introduce a novel variant of GNN to encode the input graph into latent cluster memberships. The introduced GNN, which is referred to as Heatts, uses Taylor series approximation towards the fast computation of heat kernels [5] and shows better *low pass* characteristics than Graph Convolutional Networks (GCN).

The contributions of this work are summarized as follows:

- We introduce DGVAE, which uses the Dirichlet distributions as priors on the latent variables and the latent variables are graph cluster memberships.

- We show the reconstruction term of ELBO in DGVAE can be interpreted as the spectral relaxed *balanced graph cut*.

- We identify *low pass* characteristics in the encoding stage of DGVAE and propose a novel variant of GNN based on Taylor series approximation towards heat kernels.

This paper is organized as follows. Section 2 gives preliminaries about *balanced graph cut* and GVAEs. Section 3 describes the design of DGVAE. Section 4 interprets the reconstruction term in DGVAE as *balanced graph cut*. Section 5 presents Heatts. We report the experimental results in Section 6 and discuss related work in Section 7. Finally, Section 8 concludes the paper.

## 2 Preliminaries

Formally, we are given an undirected, unweighted graph $G = (V, A, X)$. $V$ is the node set and $N = |V|$ denotes the number of nodes. The adjacency matrix $A \in \mathbb{R}^{N \times N}$ represents the graph structure. The feature matrix $X \in \mathbb{R}^{N \times d}$ represents the node attributes. Our goal is to learn an encoder and a decoder to map between the space of graph $G$ and their latent factors $Z \in \mathbb{R}^{N \times K}$.

### 2.1 Balanced graph cut

*Balanced graph cut*, i.e., *ratio cut* [30] and *normalized cut* [25], are widely used criteria for graph clustering. Formally, *graph cut* is defined as,

$$\frac{1}{K} \sum_k cut(V_k, \overline{V_k}), \tag{1}$$

where $V_k$ is the node set assigned to cluster $k$, $\overline{V_k} = V \setminus V_k$, $cut(V_k, \overline{V_k}) = \sum_{i \in V_k, j \in \overline{V_k}} A_{ij}$ and it calculates the number of edges with one end point inside cluster $k$ and the other in the rest of the graph. Suppose we are given a cluster indicator $C \in \{0, 1\}^{N \times K}$, $C_{ik} = 1$ represents node $i$ belongs to cluster $k$ and 0 otherwise. Similar to [28], we can re-write the *graph cut* as,

$$\frac{1}{K} \sum_k (C_{:,k}^\top D C_{:,k} - C_{:,k}^\top A C_{:,k}) = \frac{1}{K} \operatorname{Tr}(C^\top L C), \tag{2}$$

in which $C_{:,k}$ is the $k$-th column vector, $D$ and $L$ are the degree and Laplacian matrices for matrix $A$, respectively. $C_{:,k}^\top D C_{:,k}$ stands for the number of edges with at least one end point in $V_k$ and $C_{:,k}^\top A C_{:,k}$ counts the number of edges within cluster $k$. Unfortunately, *graph cut* favors imbalanced clustering [30, 28]. Thus, we utilize *ratio cut* [30], which is defined as,

$$\frac{1}{K} \sum_k \frac{C_{:,k}^\top L C_{:,k}}{|V_k|}, \tag{3}$$

where $|V_k| = C_{:,k}^\top C_{:,k}$ counts the number of nodes within cluster $k$. In particular, the minimum of the function $\sum_{k=1}^{K}(1/|V_k|)$ is achieved if all $|V_k|$ $(k = 1, \ldots, K)$ are the same [28]. Unfortunately, introducing this balancing condition $|V_k|$ makes *ratio cut* NP-hard (a detailed discussion can be found in [29]). In the literature, existing approaches rely on spectral relaxation [25, 9] or greedy algorithms to solve the cut problem. The most famous approach is *spectral clustering* [28], which removes the discrete constraint of cluster indicators. Due to its generality and rich theoretical foundation, *spectral clustering* has become one of the major clustering methods [2]. The algorithm of *spectral clustering* requires the computation of the first $K$ eigenvectors of the Laplacian matrix $L$, which corresponds to the smallest $K$ eigenvalues, or **low pass** for short. Denoting $\{\lambda_i\}_{i=1}^{N}$ as the eigenvalues of the (normalized) Laplacian matrix and $\lambda_K$ as the threshold, the ideal *low pass* filter $g_{\mathrm{id}}(\lambda_i)$ required by *spectral clustering* can be defined as,

$$g_{\mathrm{id}}(\lambda_i) = \begin{cases} 1 & \lambda_i \leq \lambda_K \\ 0 & \lambda_i > \lambda_K. \end{cases} \tag{4}$$

## 2.2 GNNs and VAEs

In typical GNNs and VAEs (GVAEs) such as VGAE [19] and Graphite [8], the encoder is defined as a variational posterior $q_\phi(Z|G)$ parameterized by GNNs, and the decoder is defined by a generative distribution $p_\theta(A|Z)$, where $\phi$ and $\theta$ are the corresponding parameters. Usually there is a prior distribution $p(Z)$ acting as a regularization for $q_\phi(Z|G)$. The whole framework is trained by maximizing Evidence Lower Bound (ELBO),

$$\mathcal{L}_{\mathrm{ELBO}}(\phi, \theta; G) = -\mathrm{KL}(q_\phi(Z|G)||p(Z)) + \mathbb{E}_{q_\phi(Z/G)} \log p_\theta(A|Z), \tag{5}$$

where the Kullback-Leibler divergence is defined as $\mathrm{KL}(P||Q) = \sum_j P_j \log\left(\frac{P_j}{Q_j}\right)$. The second part is the reconstruction term, which is used to guarantee the similarity between the generated structure and the input structure.

In the encoding stage of GVAEs, many studies [19, 21, 8] use GNNs to encode the input graph into node-wise latent factors. Most of them focus on deriving latent isotropic Gaussian distributions. Recently some studies [31, 23] have shown that GCN [18], one of the popular GNNs, is in essence a linear approximation to *low pass* filters in the spectral domain. Specifically, Wu et al. [31] show that GCN is a linear spectral filter with $g_c(\lambda_i) = 1 - \lambda_i$, which deviates *low pass* characteristics defined in Eq. 4 as $g_c(\lambda_i)$ becomes negative when $\lambda_i > 1$.

# 3 Dirichlet graph variational autoencoder

In this section, we present Dirichlet Graph Variational Autoencoder (DGVAE). Our primary idea is to replace Gaussian variables by the Dirichlet distributions in latent modeling of VAEs, such that the latent factors can be adopted to describe graph cluster memberships. It makes the graph generation process analogous to text generation (see LDA [1]), i.e., a graph (document) first samples clusters (topics), and then draws nodes (words) conditioned on the chosen clusters (topics).

**Encoder**  We follow VGAE [19] and Graphite [8] by using the mean field approximation to define the variational family,

$$q_\phi(Z|A, X) = \prod_{i=1}^{N} q_{\phi_i}(z_i|A, X). \tag{6}$$

As discussed, our variational marginals $q_{\phi_i}(z_i|A, X)$ are assumed to follow the Dirichlet distributions to make the latent factors interpretable. However, directly approaching the Dirichlet distributions would make the *reparameterization trick* hard to apply. To this end, we resolve this issue by using Laplace approximation [12, 27], in which the main idea is to approximate the Dirichlet distributions with the logistic normal distribution. Formally, the parameters for the variational marginals $q_{\phi_i}(z_i|A, X)$ are specified by a multi-layer GNN,

$$\mu^0, \sigma^0 = \mathrm{GNN}_\phi(A, X), \tag{7}$$

where $\mu^0 \in \mathbb{R}^K$ and $\sigma^0 \in \mathbb{R}^K$ are the vector of means and variances for the normal distributions. Following [12, 27], we can approximate the Dirichlet distributions $q_\phi(z_i|G)$ thereafter by sampling

$\epsilon \sim \mathcal{N}(0, I)$ and compute

$$z_i = \text{softmax}(\mu^0 + (\Sigma^0)^{1/2}\epsilon), \tag{8}$$

where $\Sigma^0 = \text{diag}(\sigma^0)$ returns a square diagonal matrix with the elements of vector $\sigma^0$ on the main diagonal, and $Z = \{z_i\}_{i=1}^N$. Note that $Z$ coincides with graph cluster memberships $C$, i.e., spectral relaxed cluster indicators in Section 2.1.

**KL divergence** We follow the idea in Laplace approximation [12] by rewriting $p(z_i) = Dir(\alpha)$ as the logistic normal distribution with mean $\mu^1$ and covariance matrix $\Sigma^1$,

$$\mu_k^1 = \log \alpha_k - \frac{1}{K}\sum_i \log \alpha_i, \tag{9}$$

$$\Sigma_{kk}^1 = \frac{1}{\alpha_k}(1 - \frac{2}{K}) + \frac{1}{K^2}\sum_i \frac{1}{\alpha_i}. \tag{10}$$

Now the KL divergence can be computed between two logistic normal distributions as,

$$\text{KL}(q_\phi(z_i|G)||p(z_i)) = \frac{1}{2}\{\text{Tr}((\Sigma^1)^{-1}\Sigma^0) + (\mu^1 - \mu^0)^\top (\Sigma^1)^{-1}(\mu^1 - \mu^0) - K + \log \frac{|\Sigma^1|}{|\Sigma^0|}\}. \tag{11}$$

**Decoder** Similar to VGAE [19], we approximate $p(A|Z)$ by,

$$p(A|Z) = \xi \prod_{A_{ij}=1} \exp f(C_i, C_j) \prod_{A_{ij}=0} \exp\{1 - f(C_i, C_j)\}, \tag{12}$$

where $f(\cdot, \cdot)$ denotes a distance metric, e.g., $f(C_i, C_j) = C_i^\top C_j$ or $f(C_i, C_j) = 1 - \text{MSE}(C_i, C_j)$ and $\text{MSE}(\cdot, \cdot)$ represents mean squared error. $\xi$ is a re-scaling term to ensure $p(A|Z) \in [0, 1]$.

**Component collapsing** Previous work [17] has observed that VAEs training often suffers from a particular local optimum known as *component collapsing*, in which the model reaches close to the prior belief and ignores the latent factors. This phenomenon becomes even more severe for modeling Dirichlet priors, as a good optimization of Dirichlet KL-divergence often results in a bad reconstruction term [27, 3].

We resolve this issue by aggressively optimizing the reconstruction term with more updates [11]. Specifically, we consider the optimization of Dirichlet KL-divergence and reconstruction term is imbalanced. In each iteration, we train the whole framework by separating these two terms and specializing an inner loop for updating the reconstruction term. Different from methods that modify the training objective [13, 36], our training dynamics can hold the standard ELBO tightly.

## 4 Reconstruction term as balanced graph cut

**Claim 4.1** *Maximizing the reconstruction term of DGVAE is equivalent to minimizing the spectral relaxed graph cut and a regularization that encourages balanced cluster size.*

To prove Claim 4.1, we ignore the constant and re-write the reconstruction term,

$$\log p(A|Z) \approx 2 \sum_{A_{ij}=1} f(C_i, C_j) - \sum_{A_{ij}} f(C_i, C_j), \tag{13}$$

Consider $f(C_i, C_j) = 1 - \text{MSE}(C_i, C_j)$, then the first component can be re-written as,

$$\sum_{A_{ij}=1} f(C_i, C_j) = m - \sum_{i,j}^N A_{ij}\text{MSE}(C_i, C_j)$$

$$= m - \frac{2}{K}\text{Tr}(C^\top LC), \tag{14}$$

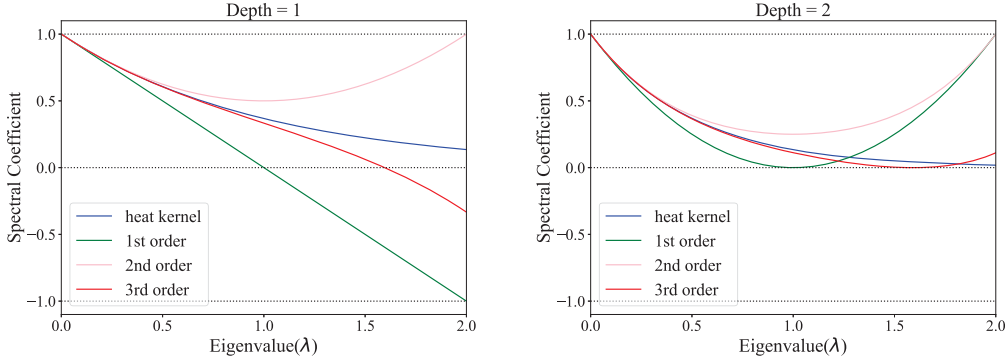

Figure 1: Spectral coefficients for low order Taylor Series approximations w.r.t. heat kernel with $s = 1$, depth denotes the number of layers.

where $m$ is the number of edges in the graph. Eq.14 shows that maximizing the first component of the reconstruction term is equivalent to minimizing spectral relaxed *graph cut*. The second component of the reconstruction term is,

$$\sum_{A_{ij}} f(C_i, C_j) = \sum_{A_{ij}} \{1 - \frac{1}{K} \sum_k (C_{ik} - C_{jk})^2\}. \tag{15}$$

Considering $\sum_{A_{ij}} \sum_{k=1}^{K} (C_{ik} - C_{jk})^2$, we aim at exploring its regularization interpretation in the graph cut problem. Here, the soft membership matrix satisfies $C \in \mathbb{R}^{N \times K}, C_i \in \Delta_{K-1}$ where $C_i$ is the $i$-th row of matrix $C$ and $\Delta_{K-1}$ is the $(K-1)$-simplex. From the statistical viewpoint, we can regard row $C_i$ in the matrix as an independent sample drawn from the posterior Dirichlet distributions, which exhibits the following characteristics,

$$C_i \sim \text{Dir}(\beta), \beta \in \mathbb{R}_+^K.$$

From this perspective, the following lemma is derived to conclude the proof of Claim 4.1.

**Lemma 4.1** *The regularization term in Eq. 15 is equivalent to maximizing the sample variance of $Dir(\beta)$ when N goes to infinity, whose optimum is achieved when all $\beta_k$ are equal and $\sum_k \beta_k \to 0$.*

Please refer to Appendix A for the proof. Lemma 4.1 shows the regularization encourages the uniform distribution of cluster memberships over $K$ clusters. Thus, it is reasonable to conclude that this regularization will help us to enforce a balanced graph cluster size.

## 5 Taylor series approximation for heat kernels

Motivated by the low pass characteristics in *balanced graph cut*, we propose a new variant of GNN to encode the input graph into cluster memberships.

A straightforward requirement for the new variant is that it should retain the low eigenvectors and drop the high eigenvectors of the Laplace matrix $L$, or *low pass* in Eq. 4. Another constraint is that it should admit fast algorithms and does not involve explicit eigendecomposition of $L$, as eigendecomposition is prohibitively expensive for large graphs [18]. In this regard, GCN uses spectral graph theory to learn such a low pass graph filter. However, the problem of GCN is that it quickly shrinks high eigenvectors at the expense of reducing useful low eigenvectors [31]. In the next, we introduce a new variant of GNN named Heatts, which utilizes the Taylor series for the fast computation of heat kernels and has better low pass characteristics than GCN.

Consider the heat kernel $g_s(\lambda) = e^{-s\lambda}$ [5], where $s > 0$ is a scaling hyperparameter. The spectral graph convolutions on a signal $x \in \mathbb{R}^N$ is defined as,

$$g_s * x = U \text{diag}(g_s(\lambda_1), \dots, g_s(\lambda_N)) U^\top x = U g_s(\Lambda) U^\top x, \tag{16}$$

where $U$ is the eigenvector matrix of the (normalized) graph Laplacian $L = U\Lambda U^\top$. Then, we apply Taylor series approximation on $g_s(\Lambda)$ and get a fast algorithm as below,

$$g_s * x = U \sum_n \frac{(-1)^n}{n!} s^n \Lambda^n U^\top x = \sum_n \frac{(-1)^n}{n!} s^n L^n x. \tag{17}$$

Table 1: Test graph generation comparison of different methods

| | Erdos-Renyi | | Ego | | Regular | | Geometric | | Power Law | | Barabasi-Albert | |
|---|---|---|---|---|---|---|---|---|---|---|---|---|
| | NLL | RMSE | NLL | RMSE | NLL | RMSE | NLL | RMSE | NLL | RMSE | NLL | RMSE |
| GAE | 0.647 | 0.535 | 0.356 | 0.346 | 0.523 | 0.455 | 0.583 | 0.370 | 0.580 | 0.401 | 0.553 | 0.428 |
| VGAE | 1.010 | 0.609 | 0.917 | 0.492 | 0.914 | 0.452 | 0.813 | 0.524 | 0.901 | 0.485 | 0.894 | 0.501 |
| Graphite-AE | 0.678 | 0.529 | 0.370 | 0.333 | 0.526 | 0.395 | 0.851 | 0.385 | 0.541 | 0.399 | 0.557 | 0.390 |
| Graphite-VAE | 1.087 | 0.602 | 0.896 | 0.496 | 0.983 | 0.474 | 0.846 | 0.536 | 0.938 | 0.466 | 0.925 | 0.482 |
| Abl-AE | 0.646 | 0.530 | 0.349 | 0.400 | 0.497 | 0.429 | 0.472 | 0.350 | 0.536 | 0.419 | 0.536 | 0.383 |
| Abl-VAE | 0.760 | 0.512 | 0.541 | 0.445 | 0.601 | 0.454 | 0.682 | 0.475 | 0.638 | 0.395 | 0.678 | 0.430 |
| DGAE | **0.239** | **0.186** | **0.250** | **0.231** | **0.305** | **0.282** | **0.406** | **0.182** | **0.383** | 0.415 | **0.308** | 0.214 |
| DGVAE | 0.286 | 0.249 | 0.436 | 0.274 | 0.516 | 0.340 | 0.537 | 0.233 | 0.519 | **0.255** | 0.346 | **0.194** |

Usually truncated low order approximation is used in practice with $n \leq 3$ [4]. Here the first order approximation is exactly GCN [18] with $g_c(\lambda_i) = 1 - \lambda_i$ when $s = 1$. Compared with the first order approximation, the third order approximation can retain more useful low eigenvectors while yielding less negative spectral coefficients when $\lambda_i > 1$. Moreover, applying multiple layers of the third order approximation can eliminate the effect of negative coefficients, which neglects trivial solutions of *renormalization trick* used in GCN [18]. In Figure 1, we plot this low order approximation with $s = 1$. The eigenvalue range for normalized Laplacian is $\lambda_i \in [0, 2]$ [31]. As Figure 1 shows, the third order approximation (in red) acts quite similar to heat kernel (in blue) when the model depth = 2. More importantly, it retains more useful low eigenvectors and drops more high eigenvectors than GCN (in green). We shall further discuss this point in Section 5.1.

Meanwhile, a feature transformation is applied to the feature matrix $X$. From another viewpoint, Heatts can be understood as a special case of a simple differentiable message-passing framework [6],

$$\text{message function} : M^l = \sum_{n=0}^{3} \frac{(-1)^n}{n!} s^n L^n H^l, \tag{18}$$

$$\text{vertex update function} : H^{l+1} = ReLU(M^l W), \tag{19}$$

where $H^0 = X$. $W$ is the parameter set to be learned.

## 5.1 Heatts and related GNNs

We first show Heatts has better *low pass* characteristics than GCN [18].

Given the eigenvalue range for normalized Laplacian is $[0, 2]$, the distance between an arbitrary spectral filter $g(\lambda)$ and the ideal *low pass* filter $g_{\text{id}}(\lambda)$ in Eq. 4 is defined as,

$$\text{Distance}(g, g_{\text{id}}) = \int_0^{\lambda_K} (1 - g(\lambda))^2 d\lambda + \int_{\lambda_K}^{2} (0 - g(\lambda))^2 d\lambda. \tag{20}$$

Intuitively, this definition computes the squared Euclidean distance between $g(\cdot)$ and $g_{\text{id}}(\cdot)$.

**Proposition 5.1** *The distance between Heatts $g_s(\lambda)$ and the ideal* low pass *filter $g_{\text{id}}(\lambda)$ is always smaller than that between GCN $g_c(\lambda)$ and $g_{\text{id}}(\lambda)$.*

Please refer to Appendix B for the proof. We then contrast the difference between Heatts and other related GNNs including Cheby-GCN [4], GraphHeat [33].

**Heatts vs. Cheby-GCN**   Cheby-GCN [4] uses parameterized Kth-order coefficients to approximate Chebyshev polynomials, which suffers from the problem of overfitting on local neighborhood structures for graphs [18]. Heatts can be considered as Chebyshev polynomials with a specific set of coefficients determined by Taylor series approximation to heat kernel.

**Heatts vs. GraphHeat**   GraphHeat uses parameterized Chebyshev polynomials to approximate heat kernel, and the learned parameters can resemble arbitrary filters. On the contrary, Heatts is an explicit Taylor series approximation towards heat kernel.

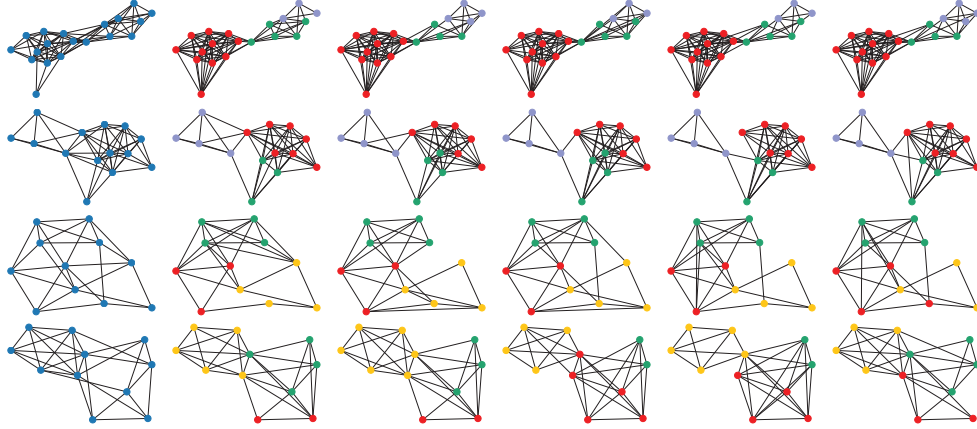

Figure 2: Left one in blue: the input graphs. Right five in colors: graph samples generated by DGVAE, where colors indicate latent cluster memberships with $K = 3$.

# 6 Experiments

## 6.1 Graph generation

**Data and baselines**    We follow Graphite [8] and create data sets from six graph families with fixed, known generative processes, to evaluate the performance of DGVAE on graph generation. For more detailed settings, please refer to [8]. We compare with GAE/VGAE [19] and Graphite-AE/VAE [8]. We denote our framework using GCN [18] in the encoders as Abl-AE/VAE.

**Setup**    For DGVAE/DGAE, we use the same network architecture through all the experiments. We train the model using minibatch based Adam optimizer. We train for 200 iterations with a learning rate of 0.01. The output dimension of the first hidden layer is 32 and that of the second-layer ($K$) is 16. The Dirichlet prior is set to be 0.01 for all dimensions if not specified otherwise. For Heatts, we let $s = 1$ for all experiments.

**Results**    The negative log-likelihood (NLL) and root mean square error (RMSE) on a test set of instances are shown in Table 1. Both DGVAE and DGAE outperform their competitors significantly on all data sets, indicating the effectiveness of DGVAE and DGVE. As an ablation study, when replacing Heatts with GCN [18], the performance is just comparable to baselines, and worse than DGVAE, which shows the superiority of Heatts.

**Component collapsing**    We evaluate how the training dynamics affects the model performance. As shown in Figure 3, this training dynamics can significantly relieve the posterior collapse problem.

**Visualization**    We plot four graphs and their samples generated by DGVAE in Figure 2 with latent cluster dimension $K = 3$. We let the number of edges for graph samples equal with the number for the input graphs. As we have analyzed, most cluster members are connected and uniformly distributed over the three clusters, indicating our model encourages a balanced graph cluster size.

## 6.2 Graph clustering

**Data and baselines**    We use three benchmark data sets, i.e., Pubmed, Citeseer [24] and Wiki [34]. Statistics of the data sets can be found in Table 3 in Appendix. As baselines, we compare against (1) Spectral Clustering (SC) [28], which only takes the node adjacency matrix as affinity matrix; (2) Node2vec [7] + Kmeans (N2v&K), which first uses Node2vec to derive node embeddings and then utilizes K-means to generate cluster results; (3) GAE [19] + Kmeans (GAE&K); and (4) VGAE [19] + Kmeans (VGAE&K). We denote our framework using GCN [18] in the encoders as Abl-AE/VAE.

**Setup**    For DGVAE/DGAE, we adopt the same settings as in the experiment of graph generation except that the output dimension of the second-layer equals to the number of clusters.

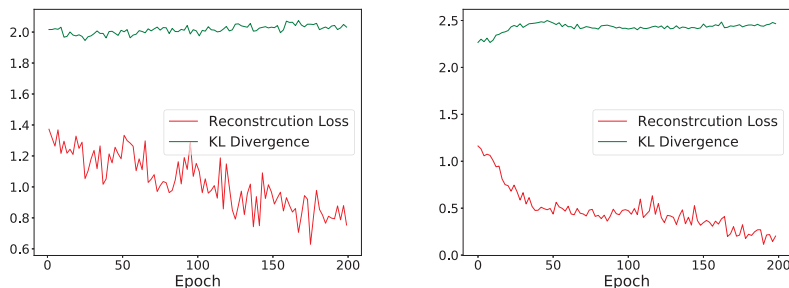

Figure 3: The training behaviour on Erdos Renyi graphs. Left: without training dynamics; right: with training dynamics.

Table 2: Cluster performance comparison of different methods

|  | Pubmed | | | Citeseer | | | Wiki | | |
|---|---|---|---|---|---|---|---|---|---|
|  | ACC (%) | NMI (%) | F1 (%) | ACC (%) | NMI (%) | F1 (%) | ACC (%) | NMI (%) | F1 (%) |
| SC | 58.3 ±0.5 | 19.0 ±0.6 | 43.2 ±0.4 | 23.9 ±1.4 | 5.9 ±3.5 | 29.5 ±2.6 | 23.6 ±3.7 | 19.3 ±3.2 | 17.3 ±2.5 |
| N2v&K | 67.7 ±1.2 | **29.5 ±1.3** | 66.3 ±1.1 | 41.3 ±1.1 | 16.7 ±0.8 | 39.5 ±1.3 | 34.9 ±1.8 | 31.1 ±2.2 | **30.3 ±1.3** |
| GAE&K | 64.2 ±1.9 | 24.0 ±1.5 | 64.4 ±1.1 | 41.2 ±0.9 | 20.8 ±1.2 | 40.1 ±1.2 | 25.1 ±1.9 | 26.7 ±1.7 | 19.8 ±1.8 |
| VGAE&K | 62.0 ±3.0 | 20.4 ±1.7 | 62.5 ±2.8 | 43.4 ±3.3 | 22.7 ±0.9 | 41.8 ±3.0 | 32.2 ±2.1 | 30.2 ±2.8 | 29.3 ±2.2 |
| Abl-AE | 66.3 ±1.4 | 25.7 ±1.9 | 66.2 ±1.7 | 46.1 ±1.9 | 24.3 ±2.2 | 40.1 ±2.2 | 38.1 ±2.3 | 33.8 ±1.6 | 24.7 ±2.1 |
| Abl-VAE | 62.6 ±2.1 | 24.2 ±2.7 | 61.4 ±2.5 | 40.2 ±2.7 | 16.1 ±2.2 | 38.5 ±2.4 | 36.2 ±2.3 | 31.4 ±1.4 | 25.9 ±2.7 |
| DGAE | **68.4 ±1.9** | 28.8 ±2.1 | **67.3 ±2.3** | **51.3 ±2.1** | **27.2 ±1.5** | **49.4 ±1.8** | **38.9 ±1.8** | **36.9 ±1.5** | 27.7 ±2.3 |
| DGVAE | 64.9 ±2.0 | 25.8 ±2.5 | 66.5 ±2.2 | 44.9 ±2.8 | 19.4 ±2.7 | 41.9 ±3.1 | 37.5 ±3.3 | 31.7 ±2.6 | 28.7 ±1.9 |

**Results** The clustering accuracy (ACC), normalized mutual information (NMI) and macro F1 score (F1) are shown in Table 2. Both DGVAE and DGAE outperform their competitors on most data sets. As DGVAE/DGAE does not rely on K-means to derive cluster memberships, this cluster performance indicates the effectiveness of our framework on graph clustering tasks. As an ablation study, when replacing Heatts with GCN [18], the performance is just comparable to baselines, and worse than DGVAE, which again proves the superiority of Heatts over GCN.

# 7   Related work

**Dirichlet VAEs** Previous studies [3, 27, 32, 15] on VAEs have enabled the usage of the Dirichlet distributions as priors, and most of them are applied in the text domain. In these practices, there are two commonly observed difficulties: (1) *reparameterization* trick is problematic when the Dirichlet distributions are applied [27], and (2) component collapsing, in which the model reaches close to the prior belief [17]. To tackle these two issues, Srivastava and Sutton [27] resolve the former by softmax Laplace approximation [12] and the latter by stacking training strategies, i.e., higher learning rate, batch normalization and dropout. Joo et al. [15] use inverse Gamma approximation to address the former issue and argue there is no component collapsing in their modeling. Burkhardt and Kramer [3] apply rejection sampling variational inference for the former and propose sparse Dirichlet VAEs to address the latter. Some other methods include Weibull distribution approximation [35] and Dirichlet stick-breaking priors [22].

**Low pass characteristics in GNN** In the literature, most variants of GNNs, e.g., Cheby-GCN [4] and GCN [18], exploit spectral graph convolutions [26] and Chebyshev polynomials [10] to retain useful information and avoid explicitly eigendecomposition of graph Laplacian. Recent studies [31, 23, 5] have noticed that GNN acts as a *low pass* filter in spectral domain and retains smoothed node representations, thus the subsequent classification tasks can be simplified [20]. However, it is not clear why *low pass* filters should be used in the encoding stage of VAEs, or what intrinsic information *low pass* filters have captured w.r.t. the whole graph. In this work, by connecting DGVAE and the spectral relaxed *balanced graph cut*, we show that *low pass* filters can be used in the encoder of DGVAE, as it can transform the input graph into latent cluster memberships, similar to the role of *spectral clustering* in *balanced graph cut*.

# 8 Conclusion

In this paper, we present DGVAE, a graph variational generative model with Dirichlet latent variables. We show the latent factors of DGVAE can be understood as cluster memberships and the reconstruction term connects with spectral relaxed balanced graph cut. Motivated by *low pass* characteristics in *balanced graph cut*, we propose Heatts, a new variant of GNN, which utilizes Taylor series for the fast computation of heat kernels and admits better *low pass* characteristics than GCN. The effectiveness of DGVAE is validated on graph generation and graph clustering tasks.

## Broader Impact

This work connects VAEs based graph generation and traditional graph research topic — balanced graph cut. As a sequence, researchers in drug design or molecule generation may benefit from this research, since the interpretation of deep learning based graph generation is worthwhile to be further explored.

## Acknowledgments and Disclosure of Funding

The work was supported by grants from the Research Grant Council of the Hong Kong Special Administrative Region, China [Project No.: CUHK 14205618], Tencent AI Lab RhinoBird Focused Research Program GF202005, and NSFC Grant No. U1936205.

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
