[Supplementary Material]

# A Proof of Lemma 4.2

The regularization is:

$$
\begin{aligned}
\sum_{i=1}^{N}\sum_{j=1}^{N}\|C_i - C_j\|_2^2 &= \sum_{i=1}^{N}\sum_{j=1}^{N}(C_i^T C_i - 2C_i^T C_j + C_i^T C_i) \\
&= 2N\sum_{i=1}^{N} C_i^T C_i - 2\sum_{i=1}^{N}\sum_{j=1}^{N} C_i^T C_j \\
&= 2N\sum_{i=1}^{N} C_i^T (C_i - \frac{1}{N}\sum_{j=1}^{N} C_j) = 2N\sum_{i=1}^{N} C_i^T (C_i - \bar{C}) \\
&\stackrel{(*)}{=} 2N\sum_{i=1}^{N} \|C_i - \bar{C}\|_2^2,
\end{aligned}
\tag{21}
$$

where $(*)$ derives from the equality $\sum_{i=1}^{N} \bar{C}^T (C_i - \bar{C}) = 0$. Hence, the regularization is used to maximize the sample variance. Assume that only samples are accessible to the target distribution $\mathrm{Dir}(\beta)$, we consider the variance instead (i.e., $\mathbb{E}_{C_i \sim \mathrm{Dir}(\beta)}[\sum_{i=1}^{N} \|C_i - \bar{C}\|_2^2]$). To simplify the notation (i.e., ignore the constant),

$$
\mathbb{E}_{x \sim \mathrm{Dir}(\beta)}[(x - \mathbb{E}[x])^T (x - \mathbb{E}[x])] = \sum_{k=1}^{K} \mathrm{Var}(x_k) = \sum_{k=1}^{K} \frac{\beta_k(\beta_0 - \beta_k)}{\beta_0^2(\beta_0 + 1)}.
\tag{22}
$$

Here, we have $\beta_0 = \sum_{k=1}^{K} \beta_k$. We want to investigate the effect of adding this regularization w.r.t. to parameter $\beta$. Alternatively, we consider the optimization problem as below,

$$
\begin{aligned}
\max_{\beta} &\sum_{k=1}^{K} \frac{\beta_k(\beta_0 - \beta_k)}{\beta_0^2(\beta_0 + 1)} \\
\text{s.t. } &\beta_k \geq 0, \forall k \in [K].
\end{aligned}
\tag{23}
$$

Then, we have $t^* = \sum_{k=1}^{K} \beta_k^*$ where $\beta^*$ is the optimal point. We take a step towards the following convex problem,

$$
\begin{aligned}
\min_{\beta} &-\sum_{k=1}^{K} \beta_k(t^* - \beta_k) \\
\text{s.t. } &\sum_{k=1}^{K} \beta_k = t^*, \\
&\beta_k \geq 0, \forall k \in [K].
\end{aligned}
\tag{24}
$$

As the slater condition holds, KKT condition is necessary and sufficient. The so-called augmented Lagrangian function is

$$
L(\beta, \nu, \pi) = -\sum_{k=1}^{K} \beta_k(t^* - \beta_k) + \nu(\sum_{k=1}^{K} \beta_k - t^*) - \sum_{k=1}^{K} \pi_k \beta_k.
$$

The KKT condition is

$$
\begin{cases}
-t^* + 2\beta_k + \nu - \pi_k = 0, \forall k \in [K] \\
\sum_{k=1}^{K} \beta_k - t^* = 0, \\
\pi_k \beta_k = 0, \beta_k \geq 0, \pi_k \geq 0 \ \forall k \in [K]
\end{cases}
\tag{25}
$$

Consider the case $\pi_k = 0, \beta_k > 0$, we have $\beta_k^* = \frac{t^*}{K}, \nu^* = \frac{K-2}{K} t^*$. We come back to the original problem,

$$
\sum_{k=1}^{K} \frac{\beta_k(\beta_0 - \beta_k)}{\beta_0^2(\beta_0 + 1)} = \frac{K-1}{Kt^*}.
$$

Overall, maximizing this component enforces $t^* \to 0$ and all equal parameters for the Dirichlet distributions.

# B Proof of Proposition 5.1

The distance between an arbitrary spectral filter $g(\lambda)$ and the ideal *low pass* filter $g_{\mathrm{id}}(\lambda)$ in Eq. 4 is defined as,

$$
\mathrm{Distance}(g, g_{\mathrm{id}}) = \int_0^{\lambda_K} (1 - g(\lambda))^2 d\lambda + \int_{\lambda_K}^2 (0 - g(\lambda))^2 d\lambda.
\tag{26}
$$

Intuitively, this definition computes the squared Euclidean distance between $g(\cdot)$ and $g_{\text{id}}(\cdot)$. Thus, the distance between GCN $g_c(\cdot)$ and the ideal low pass $g_{\text{id}}(\cdot)$ is:

$$\text{Distance}(g_c, g_{\text{id}}) \quad = \int_0^{\lambda_K} (1 - (1 - \lambda))^2 d\lambda + \int_{\lambda_K}^2 (0 - (1 - \lambda))^2 d\lambda$$
$$= \lambda_K^2 - \lambda_K + \tfrac{2}{3} \tag{27}$$

The distance between Heatts $g_s(\cdot)$ and the ideal low pass $g_{\text{id}}(\cdot)$:

$$\text{Distance}(g_s, g_{\text{id}}) \quad = \int_0^{\lambda_K} (-s\lambda + \tfrac{1}{2}s^2\lambda^2 - \tfrac{1}{6}s^3\lambda^3)^2 d\lambda + \int_{\lambda_K}^2 (1 - s\lambda + \tfrac{1}{2}s^2\lambda^2 - \tfrac{1}{6}s^3\lambda^3)^2 d\lambda \tag{28}$$

Our purpose is to derive value range of $s$ such that $\text{Distance}(g_s, g_{\text{id}})$ is always smaller than $\text{Distance}(g_c, g_{\text{id}})$.

$$\text{Distance}(g_s, g_{\text{id}}) - \text{Distance}(g_c, g_{\text{id}}) \geq 0 \tag{29}$$

The solution is $0.672 \leq s \leq 1.321$

As shown in Figure 4, Heatts is always closer to the ideal low pass $g_{\text{id}}(\cdot)$ when $s \in [0.672, 1.321]$.

Table 3: Statistics of data sets used in graph clustering

| Data | Nodes | Edges | Classes | features |
|------|-------|-------|---------|----------|
| Pubmed | 19,717 | 44,338 | 3 | 500 |
| Citeseer | 3,327 | 4,732 | 6 | 3,703 |
| Wiki | 2,405 | 17,981 | 17 | 4,973 |

Figure 4: The distance between spectral filters and the ideal low pass