[Reviews · NeurIPS 2020]

Review 1

Summary and Contributions: The paper proposes a Dirichlet graph variational autoencoder, an instance of a variational autoencoder in which the input graph is encoded into Dirichlet-distributed latent variables. As a consequence, they can be interpreted as cluster memberships, similar to topic model VAEs for text generation. The paper also establishes a connection between a term in the ELBO and an instance of graph cuts, which motivates a novel graph encoding strategy based on a Taylor approximation of heat kernels in spectral graph convolutions.

Strengths: - The paper addresses an important problem and its results, specifically regarding the semantic understanding of latent variables, could be relevant in a variety of contexts. - The proposed Heatts encoder is well motivated and analyzed. I particularly liked the theoretical comparison to GCNs in terms of their distance to the ideal low pass filter. - I enjoyed reading about the interesting relationship between elements of the ELBO and a version of graph cuts. [see below for questions about the proof, though] - The experiments show performance improvements over popular baselines in a number of relevant scenarios. Furthermore, an ablation study shows the effectiveness of the proposed Heatts encoder compared to traditional GCN-based encoding. - Visualizations of graph clusters based on latent cluster memberships serve as empirical evidence of the paper’s technical contributions.

Weaknesses: - The established relationship between spectrally relaxed graph cuts and the reconstruction term of the ELBO (claim 4.1) feels fragile; please see below for additional details. I am also confused about the implications of claim 4.1 in a more general sense: does the proof rely on a specific distribution of the latent variables? If not, does that mean the statement is also true for VGAEs, which use a very similar generative model? - Section 2 provides a good summary about graph cuts, but more intuition about the ratio cut objective and additional details about spectral clustering would have been helpful; maybe they can be added to the supplementary material. Especially the role of Eq.(4) does not become immediately clear. - Section 3 heavily builds upon results in [12], but in many cases the necessary context is missing. For example, it is not possible to understand why Eq.(8) is Dirichlet-distributed or why Eqs.(9-10) describe a logistic normal distribution without consulting external references. I encourage the authors to update this section with additional details to make it more self-contained. - The quality of the Laplace approximation proposed in section 3 is unclear. Is it possible to make a theoretical statement in terms of an error bound? - The experimental evaluation could be improved in two ways: (1) The description of the datasets is very brief, simply referring to [8] is not enough; (2) The evaluation distinguishes x-AE and x-VAE, but it is nowhere mentioned what the difference is.

Correctness: - Eq.(12) seems to differ from the decoding strategy proposed in [19]. Were these differences introduced on purpose and, if so, what is the motivation behind them? Why does Z not appear on the RHS of the equation? Is there an implicit assumption that the latent variables correspond to cluster memberships? - Claim 4.1 establishes a connection between the ELBO reconstruction term and spectrally relaxed graph cuts. However, it only considers the decoder p(A|Z) and ignores the effects of the approximate posterior in the reconstruction term. Furthermore, it is not clear for which distance metrics f claim 4.1 holds. I encourage the authors to comment on that.

Clarity: - The paper is well written and for the most part easy to follow. The structure is clear and the goals and objectives are well motivated. The notation and terminology are consistent.

Relation to Prior Work: - The introduction puts the proposed model in context with relevant prior works on graph variational autoencoders, graph cuts, and topic model VAEs for text generation. The related work section expands this discussion to Dirichlet VAEs and literature on the low pass properties of GNNs. - Since the proposed model is heavily inspired by topic model VAEs for text generation, it is not always clear which ideas are simple adaptions of earlier works and which ones are truly novel contributions. A more detailed discussion of these earlier works feels necessary.

Reproducibility: Yes

Additional Feedback: I'm willing to raise my score if my questions/concerns under 'Correctness' are addressed. Post-rebuttal comment: I thank the authors for addressing my concerns about some of the technical aspects of the paper and increase my score from (6) to (7).


Review 2

Summary and Contributions: Authors propose to extend the variational graph autoencoders by replacing Gaussian distributed latent variables with the Dirichlet distributed variables (approximated by logistic normal) such that the latent variables can be directly used to describe graph cluster memberships. The model is trained by optimising the evidence lower bound, and authors show that maximizing the reconstruction term is equivalent to minimizing the spectral graph cut and that the regularization term (KL) promotes balanced cluster sizes in the latent space. Authors report competitive results on graph generation and graph clustering, when compared to existing methods. Based on authors' ablation study much of the improved performance can be attributed to a new GNN encoder that embeds a given input graph into (latent) cluster membership. The proposed GNN uses graph convolutional neural network with a so-called heat kernel (together with a tailor approximation).

Strengths: Competitive results on several data sets when compared to previous methods. Theoretical results showing that the proposed method encourages balanced graph cuts and balanced cluster sizes. For the most part, the manuscript is well-written, and the introductory/preliminaries part is written in an educative way.

Weaknesses: While results on simulation studies seem competitive, the best results seem to be achieved with DGAE: I understood that DGAE is a non-variationally trained alternative of the proposed method, and authors need to further clarify whether/to what extend the theoretical results apply to DGAE variant of the method. The fact that the method promotes balanced cluster sizes may perhaps also be seen as a limitation, as not all data sets necessarily have equally sized clusters.

Correctness: I did not check all derivations thoroughly but they seem correct.

Clarity: For the most part, yes. I would like to see a more detailed description of Heats kernel based encode at the level of connecting it to the parameters of the variational approximation q().

Relation to Prior Work: Yes.

Reproducibility: Yes

Additional Feedback: I would like to see a more detailed description of Heats kernel based encode at the level of connecting it to the parameters of the variational approximation q(). Eq. 5: Expectation with respect to q_\phi(Z \mid G). UPDATE: upon reading authors‘ response letter, I changed my overall score = 7


Review 3

Summary and Contributions: The paper deals with learning a variational autoencoder for graph structure generation. The authors draw connections to spherical clustering and the corresponding balanced graph cut. In addition, they introduce a novel graph neural network based on heat kernels in conjunction with a Taylor series for fast computation. The methods has been evaluated on graph generation and graph clustering tasks on both artificial and real world datasets.

Strengths: * The authors introduce a three-fold contribution for learning graph-structures in a VAE setting. Especially, they propose a new GNN based on heat kernels that learns better latent representations for graph structure generation tasks and draw connections to spherical clustering. * The proposed method was extensively tested on multiple datasets and compared to multiple baseline approaches.

Weaknesses: * The authors formulate the problem that the explanation of latent factors remain unclear in the context GGNs and VAEs. To overcome these limitations, they proposed DGVAE however it is unclear to me in what sense the Dirichlet approach leads to a better explanation compared to other clustering approaches. * It would be interesting to compare your approach to more recent state-of-the-art clustering approach instead of plain k-means such as [4] * The three proposed contributions seems to be only loosely connected. For example, in the experimental evaluation one gets the impression that only Heatts is required to obtain the good results (e.g. l. 214). Therefore, I sometimes got the impression that the other contributions are nice to have however not necessary for the whole approach. It would be nice if the authors could try embed their contributions better in their story such that they are also reflected in the experiments.

Correctness: The claims of the paper seem to be correct and the empirical methodology seems to be correct.

Clarity: The paper is tightly written but the methodology is sufficiently explained. Sometimes I miss a bit the common thread throughout the paper.

Relation to Prior Work: The authors clearly discussed their method and the differences to prior work. However, I would have appreciated a discussion about archetypal analysis as it seems to be highly-related to Dirichlet-based autoencoders, e.g. [1,2,3]

Reproducibility: Yes

Additional Feedback: * How do you make sure that data points lie at the corners of your simplex. Is it is only imposed by the prior assumption and by optimizing the reconstruction term with more updates? It would be interesting to know how stable this approach is? * As a follow up question, how do you sample from our latent space and are there big holes? Because if you try to cluster the data points in the corners most parts of the remaining space should be empty? * In line 108: The authors write that the Dirichlet assumption makes the latent factors more interpretable. In what sense and how does it differ from a normal clustering? * I am wondering if it is fair to compare cluster-based methods such as k-means that look for typical observations (cluster centers) with archetypal-like constructions that seek for extreme points where all other data points are convex-combinations of such extreme points (corners of the Dirichlet simplex). =============================================================== After rebuttal: Dear authors, thank you for answering and clarifying my questions. After reading your response and the discussion with the other reviewers, I will adjust my overall score to 7.

[Author Response · NeurIPS 2020]

We sincerely thank all reviewers for their valuable comments. Our responses to the comments are listed below.

Q1. About Eq.12 and reference [19] (q1 under **Correctness**).

$$
\begin{aligned}
p(A|Z) &= \xi \prod_{A_{ij}=1} \exp(f(C_i, C_j)) \prod_{A_{ij}=0} \exp(1 - f(C_i, C_j)) \\
&\approx \prod_{A_{ij}=1} \frac{\exp(f(C_i,C_j))}{\exp(f(C_i,C_j))+\exp(1-f(C_i,C_j))} \prod_{A_{ij}=0} \frac{\exp(1-f(C_i,C_j))}{\exp(f(C_i,C_j))+\exp(1-f(C_i,C_j))} \\
&= \prod_{A_{ij}=1} \frac{1}{1+\exp(1-2f(C_i,C_j))} \prod_{A_{ij}=0} \frac{\exp(1-2f(C_i,C_j))}{1+\exp(1-2f(C_i,C_j))} \\
&= \prod_{A_{ij}=1} \sigma(C_i^\top C_j) \prod_{A_{ij}=0} (1 - \sigma(C_i^\top C_j)) \quad \text{consider } f(C_i, C_j) = \tfrac{1}{2} C_j^\top C_j + \tfrac{1}{2}
\end{aligned} \tag{1}
$$

where $\sigma(\cdot)$ is the logistic sigmoid function and the last line in the above equation is the one used in [19]. $Z$ does not
appear on the RHS of the equation, as $Z$ coincides with cluster memberships $C$ (line 117 in the paper). This point can
be verified since both $z_i \in \Delta_{K-1}$ and $C_i \in \Delta_{K-1}$ are the $(K-1)$-simplex.

Q2. About Claim 4.1 and ELBO (q2 under **Correctness**). If we understand it correctly, "the approximate posterior in
the reconstruction term" refers to the KL term in ELBO. If so, we do consider the effects of the KL term in the proof.
As in line 107 of the paper, we regard $C_i \sim \mathrm{Dir}(\beta)$, which is a Dirichlet posterior with a Dirichlet prior $\mathrm{Dir}(\alpha)$. In other
words, our proof starts with the effects of the KL term. We set the distance metric to be $f(C_i, C_j) = 1 - \mathrm{MSE}(C_i, C_j)$
(line 140 in the paper) in the proof of Claim 4.1.

Q3. Does Claim 4.1 rely on a specific distribution (q1 under **Weaknesses**)? A quick answer is yes, as our proof relies
on the fact $C_i \sim \mathrm{Dir}(\beta)$. As for Gaussian variables (the case in VGAE), we are not sure if the claim still holds.

Q4. Laplace approximation towards Dirichlet (q3 under **Weaknesses**). In section 3, we used $\mathcal{N}(z_i'; \mu^0, \sigma^0)$ to
approximate the Dirichlet distribution $\mathrm{Dir}(z_i; \phi)$, where $z_i = \mathrm{softmax}(z_i')$ is based on softmax Laplace approximation.
We agree sometimes it might be hard to follow and we shall add the details in the revision.

Q1. About Heatts and $q_\phi(\cdot)$ (q1 under **Feedback**). The parameters of the variational approximation
$q_\phi(\cdot)$ are determined by Eq.7 of the DGVAE framework. Putting Heatts into the framework, we derive $\mu^0, \sigma^0 =$
$\mathrm{Heatts}_\phi(A, X)$, as Heatts is a variant of GNN.

Q2. About DGVAE and DGAE (q1 under **Weaknesses**). We agree non-variational version performs better in most cases
than its variational one, as observed by many other references [8][19]. In our opinion, the reasons are: (1) component
collapsing, a good optimization of KL term results in a bad reconstruction term; (2) inappropriate priors, we use the
same priors through the experiments. As for the theoretical results (Claim 4.1), we think it holds for DGAE, if we
consider the Dirichlet posterior is only determined by the likelihood, i.e., without priors.

Q3. About the limitation of balanced cluster sizes (q2 under **Weaknesses**). We agree balanced cluster sizes could be
a limitation in some cases. However, Claim 4.1 is based on asymptotic analysis. Moreover, the optimization is also
affected by the Dirichlet priors, which provides our model with flexibility.

Q1. About the interpretation of Dirichlet factors (q1 under **Weaknesses**). We agree that the Dirichlet
approach does not lead to a better explanation compared to other clustering approaches, e.g., spherical k-means.
Compared to the traditional graph variational auto-encoders which focus on Gaussian distributions, Dirichlet approach
leads to a better explanation since its latent variables can be interpreted as cluster memberships.

Q2. About comparison of SOTA (q2 under **Weaknesses**). We report the experimental results of VGAE + spherical
K-means (VGAE&S), for Pubmed, VGAE&S achieves 61.2%, 16.7% and 61.1% on ACC, NMI and F1 respectively;
for Citeseer, VGAE&S gets 46.8%, 21.7% and 44.7% respectively; for Wiki, VGAE&S gets 27.8%, 24.3% and 19.5%
respectively. The results show our method outperforms VGAE&S on all metrics. In addition, we would like to highlight
that our method does not rely on any outsourcing, e.g., spherical K-means or GMM, and cluster memberships can be
derived directly from the latent Z, while previous approaches such as VGAE do not have this capacity.

Q3. About the three contributions (q3 under **Weaknesses**). Yes, we agree Heatts is the key component that leads to the
good experimental results. However, the good results are not the sole purpose of this work. More precisely, this work
wants to figure out a clear way to understand how the graphs are generated and how we can improve the design of graph
VAEs analytically. In this vein, we first describe the DGVAE framework (the 1st contribution); we then analyse the
framework and find that the optimization of the framework favors *low pass* of GNN (the 2nd contribution); based on
that, we propose Heatts. Without the 1st and 2nd contributions, the design of Heatts may reduce to trial-and-error.

Q4. About the corners of simplex (q1 under **Feedback**). Yes, it is imposed by the prior assumption and by optimizing
the reconstruction term. At the initial stage of our model (randomly initialized parameters), the data may appear
anywhere in the simplex. As the optimization goes, the Dirichlet posterior $C_i \sim \mathrm{Dir}(\beta)$ would be "dragged" towards
$\sum_k \beta_k \to 0$ (Lemma 4.2), which makes the data sampled from this distribution lie at the corners of the simplex. For
the stability, the right figure of Fig. 3 provides an intuitive way to understand the process. As shown, the learning curve
is smooth and drops quickly, indicating our approach is stable. The sampling strategy is illustrated in Eq.8.

[Meta-Review · NeurIPS 2020]

Although the validation section would benefit from a somewhat more extensive set of baselines, the paper presents a nice and apparently well-working extension to VAEs.